# Exploratory Study into the Classification Agreement between Self-Reported Age of Menarche and Calculated Maturity Offset in Adolescent Girls: A Two-Year Follow-Up Study

**DOI:** 10.3390/jfmk9030127

**Published:** 2024-07-19

**Authors:** Barry Gerber, Anita E. Pienaar

**Affiliations:** Physical Activity Sport and Recreation (PHASReC) in the Faculty of Health Science, Potchefstroom Campus, North-West University, Potchefstroom 2531, South Africa; anita.pienaar@nwu.ac.za

**Keywords:** adolescence, maturation, menarche, physical literacy, physical activity

## Abstract

Menarche is a significant pubertal event influencing girls’ participation in physical activity. As menarche is a sensitive matter, a non-invasive substitute is needed to help classify girls’ maturity status and provide physical literacy to them in this regard. The objective of this exploratory study was to investigate the classification agreement between self-reported age of menarche and calculated maturity offset in adolescent girls from South Africa by making use of various statistical methods. Fifty-eight girls, n = 13 pre- and n = 45 post-menarche (Status Quo method) aged 13.51 ± 3.51 years at baseline, were analyzed (2010–2012). Independent t-testing, cross-tabulation, Roc Curve statistics and logistic regression were used to analyze the classification agreement between markers. All four statistical methods revealed the potential to categorize different maturity groups through the maturity offset equation, although the accuracy declined with increased age. A realized power of 0.92 was found for the group in the first year of the study, with a gradual and significant decline over time. Cross-tabs showed a significant moderate predictive effectiveness (Chi-square = 0.042) during T1, closer to PHV (13.51 years) although also declining significantly with increased age (T2, 14.51 years) beyond PHV (Chi-square = 0.459). Although positive results were found, caution must be used when using maturity offset equations in different homogenic populations due to their unique growth characteristics.

## 1. Introduction

Adolescence represents a time in human life wherein various changes occur between childhood and adulthood [1]. Among girls, adolescence is characterized by a global acceleration in growth and maturation between the ages of 12 and 21 years [2,3]. Furthermore, the adolescent period is characterized by three main changes including sexual maturity, increase in stature and weight, and changes in body composition [4]. When focusing on biological development and sexual maturity of girls entering the pubertal stage, the landmarks of maturational events are the onset of puberty, peak height velocity (PHV) and menarche [5].

The pubertal phase signifies the beginning of adolescence when the gonads become functional, and consequently, females begin to develop secondary sexual characteristics [3]. This period includes girls reaching PHV at a mean age of 12 years (y) [6], and menarche about 1–2 years after PHV at an average age of 12.8 years. This mean age has been used to categorize girls as early (<11.8 y), average (11.8–13.8 y) and late (>13.8 y) developers. Menarche is one of the most remarkable morphological changes that occur during adolescence with significant consequences in physical and psychological aspects [7]. Menarche leads to diverse hormonal changes and in turn changes in body composition including increased fat deposition [8] while also affecting the psychological and cognitive development of girls necessary for coping with adulthood [8]. Furthermore, the onset of the menstrual cycles has various negative influence on girls’ PA and sport participation before and during the menstrual cycle [9]. In this regard, Jones and co-workers reported that girls indicated negative cyclical symptoms affecting their PA and sport participation, such as bloating, lower-back- and pelvic pain, sleep disturbances, headaches and clumsiness before and during their menstrual cycles. The performance of more than 50% of power-, endurance- and throwing athletes were affected negatively during the menstrual cycle [9]. In this regard, differences in biological maturation before menarche, but also because of menarche, result in growth differences and more specifically weight-related changes that in turn will affect other aspects of life such as motor- and physical skill performance [10,11], which again can influence their PA levels in the future. Therefore, girls need to be aware of these changes and effects of menarche/menstrual cycle. In this regard, one of the theoretically important elements for developing and achieving lifelong participation in PA is physical literacy (PL). PL is broadly defined as a “disposition to capitalize on our human-embodied capability wherein the individual has the motivation, confidence, physical competence, knowledge, and understanding to value and take responsibility for maintaining purposeful physical pursuits and activities throughout the life course” [12]. To manage participation in sports, most sporting codes group children by chronological age and not by biological age [13], although chronological age is of limited utility in the assessment of growth and maturation [14]. The need to assess maturation and the tempo and timing of the progress toward the mature state is, however, imperative in the study of child- and adolescent growth. Existing methodology does provide the required mechanisms to assess maturation processes before maturity offset but also regarding menarche status, although there are limitations to the available methodologies [15,16,17].

Different research methods are used to determine age of menarche including monitoring studies based on prospective and longitudinal methods where girls are examined every 3–6 months [18]. Interviews with girls or their mothers are also conducted where researchers rely on recall information about when menarche occurred. This recall method may be less valid, and the accuracy decrease with a greater time interval between menarche and the date when girls were asked to recall onset [18]). Furthermore, all girls to be evaluated by this method should be at an age where they have already reached menarche, because it is a recall of a past event and not a method that examines the current state of menarche [19]. Several studies have investigated the correlation between the actual age and the recall age of menarche and have found correlations of 0.75 after 17 years of recall and 0.60 after 39 years of recall [20]. Furthermore, Koo and Rohan found after a 2-year recall period that only 59.1% of the group could recall the correct month and year while 77.3% showed a deviation of less than 1 month of real time. Lastly, the Status Quo method is another method [4] to obtain information regarding age of menarche by only asking the girl (or the mother) about the current status of menarche; in other words, whether she had her first menstruation before or during the time of assessment. The answer in this method will be indicated by a YES or a NO, and if YES, a year and month of the first menstruation should be included [21].

Some of these methods are, however, costly, time-consuming and invasive to girls’ privacy [22]. Because of differences in the timing of the adolescent growth, more non-invasive methods are needed to classify girls into early-, average or late-maturing groups [22]. Therefore, the need for a valid, less invasive, once-off assessment of biological growth and maturation of girls that can be used in schools and field settings was considered an important matter [23].

Peak height velocity (PHV) is one milestone that can be used to determine maturity status earlier in the pubertal phase of girls. In this regard, maturity offset is a cross-sectional measure that estimates age at PHV and is expressed in years around PHV. A negative value will indicate pre-PHV and a positive value, post-PHV. Mirwald and co-workers [18] suggest, according to their original equations, that calculated maturity offset equations are accurate up to four years pre-PHV but no more than two years post-PHV, and within 8 to 16 years of age. Growth measurements and ratios that are used in these equations for prediction purposes are based on age and include measurements of stature, sitting height, sub-ischial leg length, and weight [18]. Interactions between leg length and sitting height are also used [18]. Lastly, different ratios, including weight to height, sitting height to total stature, leg length to total stature, and leg length to sitting height ratios, are calculated in these equations [18]. In newer versions of the equations, where researchers aimed to increase the feasibility of the equations, sitting height was removed from the equations [15,17]. Mirwald and co-workers, however, report a small average difference of only 0.021 ± 0.489 years for boys and girls between the predicted age and actual age of PHV. This equation can subsequently be useful in reducing the limitations of other methods in determining differences in the biological development of girls [18,24].

Several researchers have tested the above-mentioned equations [15,17,23]. Results from these studies found certain limitations including variations in body segmental lengths between ethnic groups and, furthermore, that prediction of maturity offset is dependent on chronological age and had reduced ranges of variation. Malina and co-workers [23] concluded that on average, equations over-predicted age of maturity offset among early-maturing youth and under-predict age of maturity among late-maturing youth. Various other studies also concluded that more work is still needed in this area, as it is important to simplify and calibrate equations to accurately fulfil the need for a valid, less invasive, once-off assessment of biological growth and maturation [15,16,17]. From a South African perspective, limited studies have used or tested these developed equations on a South African youth population. One study used the equation, but only on boys and only to classify boys into maturity groups, and did not focus on the agreement between actual and predicted ages of maturity [25]. Fewer research is also available worldwide including South Africa on girls in this research area.

Consequently, due to a lack of studies in SA, the effectiveness of these equations needs more investigation as a non-invasive tool to determine the state of biological maturation of girls earlier in their biological development. Such information can provide necessary information regarding different influences related to the physical development of girls to practitioners, physical educators, and to girls themselves (physical literacy). In this regard, information from these non-invasive tools can be used to improve girls’ physical literacy as such information can provide a better understanding of developing differences between girls, and more specifically biological and physical developmental differences in girls. Girls’ understanding of their current biological development and the challenges and opportunities that it creates regarding their physical development, participation in PA and sport performance can also be increased. Furthermore, this knowledge can then aid practitioners, coaches and physical educators in developing more customized training and PA programs to manage PA and optimize sport development and performance. Therefore, the aim of this exploratory study was to investigate the level of agreement of the classification of girls and to categorize them back into the same early- and average/late-maturing group based on their menarche status method and the calculated method (Status Quo, self-reported) of maturity offset (reaching of PHV), based on longitudinal data from a two-year follow-up study.

## 2. Materials and Methods

### 2.1. Study Design and Sample


**Recruitment.**


This original longitudinal study, ‘Growth and sport psychological characteristics of talented adolescents’, spanned from 2010 to 2012. The data were collected by means of anthropometric measurements as well as questionnaires. The anthropometric measurements took place 3 times annually (4 months apart in February, June and November). Baseline measurements were taken in February 2010 and the last measurements took place in November 2012, including 9 time-point measures for growth (T1–9). Overall, follow-up measures were obtained over a full two-year period. For the purpose of this study, only the anthropometric measures taken at T1 (February 2010, Grade 8), T2 (February 2011, Grade 9) and T3 (February 2012, Grade 10) will be used. For the purpose of the study, the above measurement points will be referred to as T1, T2 and T3.


**Participants.**


All Grade 8 learners, with a mean age of 13.51 ± 3.51 at baseline from one quintile 5 school (quintile 1 = low socio-economic quintile, to quintile 5 = high socio-economic) in Potchefstroom in the North West Province of South-Africa, were invited to take part in the research project. Since the school had boarding facilities, Grade 8 learners enrolled in 2010 represented learners from 46 different primary schools. The final group of girls, who completed all follow-up measurements at T3 in 2012, included 58 girls with a mean age of 15.52 years. This group were further divided into a pre- (n = 13) and post-menarche group (n = 45) according to their menstrual status at baseline in 2010 (T1). The pre-menarche group included all girls who had not reached menarche during the time of baseline assessment, and they were classified as late-developing girls, while the post-menarche group included the rest who have reached menarche at or before the time of baseline assessment, classified as average- and late-maturing girls. Over the study period, 37 girls (38%) were lost to follow-up due to various reasons including parents of children moving out of town, children moving between schools and injuries. The current study, however, complies with the results of a priori power analysis, which indicated that group sizes of 14 participants are needed to observe medium effects of differences. Due to this study being an exploratory study, the current population size was deemed sufficient for the purpose of the study.

Ethical approval was obtained for the execution of the study (NWU 00199-15-A1). Permission for the project was also obtained from the principal of the school involved. All girls and their parents received a parental and participation permission consent form containing all the information explaining the purpose and objective of the study as well as the procedure for all the tests. They also had the opportunity to ask questions at an information meeting or by contacting the primary investigator. All parents or legal guardians and participants voluntarily signed consent. Only the girls who had parental permission and who gave assent themselves were included in the study.


**Measures**



**Age at menarche.**


The age of menarche was determined by the Status Quo method [26,27] during February in 2010, 2011 and 2012 (T1, T2 and T3), which was the first measurement of each school year. The subjects had to indicate on a questionnaire by selecting YES or NO whether they have had their first menstrual cycle on or before the date of testing during this first measurement. When answered YES, they also had to indicate by recall, retrospectively, the month and year when their first menstrual cycle took place. The date of birth of each participant was also collected to determine their exact chronological age. High percentage of the 58 girls in the study (77%, n = 45, post-menarche group) already reached menarche at a mean age of 13.51 years during baseline measurements in 2010 (Grade 8). Thereafter, 91% (n = 53) of the group (which include nine of the pre-menarche group) reached this milestone during the first follow-up measurements in 2011 (Grade 9) at a mean age of 14.52 years with all (n = 57) but one subject (98%) reaching menarche at a mean age of 15.52 years during the last year of measurements at T3 in 2012 (Grade 10). For the purpose of this study, the 45 girls who had reached menarche at baseline (T1) and who were classified as post-menarcheal were compared with the 13 girls that did not reach menarche at baseline and who were classified as pre-menarcheal.

Maturity offset is calculated by means of an equation that includes cross-sectional measures from which age at PHV can be estimated. Maturity offset was calculated through a gender specific equation that was developed by Mirwald and co-workers [18] which looks as follows:

Maturity offset in girls = −9.376 + 0.0001882·× (leg length and sitting height interaction) + 0.0022 ×·(age and leg length interaction) + 0.005841 ×·(age and sitting height interaction) − 0.002658 ×·(age and weight interaction) + 0.07693 ×·(weight by height ratio). To determine interactions (e.g., leg length and sitting height interaction), the two factors are multiplied (leg length x sitting height). Ratios are calculated by dividing the first variable in the parentheses by the second variable and then multiplying the answer by 100. The score is expressed in years around PHV where a negative value indicates pre-PHV and a positive value, post-PHV. A reliability coefficient of r = 0.94, r^2^ = 0.890, and a Standard Error of Estimate of (SEE) = 0.569 are reported for the equation.

### 2.2. Anthropometric Measurements

All researchers were trained beforehand to accurately take the measurements associated with the study (Level 2 Kinanthropometrists), theory and practical training), ranging from honor’s students to senior researchers in Kinderkinetics and Sport Science. All measurements were taken in enclosed spaces to ensure the privacy of the participants and were taken by researchers of the same gender as those of the participants to ensure good clinical practice. Participants had to wear minimal clothing during measurements.

The protocol as developed by the International Society for the Advancement of Kinanthropometry (ISAK) was used for stature, sitting height and body mass measurements (Ameade & Garti, 2016; [28] and the Canadian Sports for Life protocol [Simmons, 2000, as cited by [29]] was used for arm span measurements. Stature and sitting height were measured to the nearest 0.1 cm by means of a portable Harpenden stadiometer (Harpenden Holtain Ltd., Crymych, UK) and weight to the nearest 0.1 kg with the use of an Omron BF 511 electronically calibrated scale (OMRON, Kyoto, Japan). Arm span was measured with steel tape measure to the nearest 0.1 cm [28] and leg length was calculated by subtracting the sitting height value from the stature value. For all variables, two measurements were taken, of which an average value was used. Inter-measurement reliability was calculated as 0.98.

### 2.3. Statistical Analysis

The data were analyzed by means of the “Statistica for Windows 2017” and “IBM SPSS Version 25” computer program packages. The data were analyzed descriptively by using means, standard deviations (SD) and minimum and maximum values [30]. The data were first checked for normality by means of a Shapiro–Wilk analysis and no serious deviations of normality were detected. Independent t-testing analyzed maturity offset differences between pre- and post-menarche girls, where a *p* < 0.05 level was used as the set criteria for statistical significance. The effect sizes of differences were calculated by using cut-off values as developed by Cohen, where a d-value of d > 0.3 reflects a small practical significance, a d > 0.5 reflecting a medium significance, and a d > 0.8 reflecting a large significance [31]. Cross-tabulation with a chi-square was used to determine the association between Status Quo and Maturity offset results with Cramer’s V as a measure of the association where V > 0.1 = small predictive effect, V > 0.3 = medium predictive effect, and V > 0.5 = large predictive effect [32]. SPSS was used to perform a ROC curves analysis as well as logistic regression to determine if the maturity offset formula can accurately predict maturity status. ROC curves (Receiver Operating Characteristic) were also used to determine maturity offset cut-off points for early- and late-maturing girls with an asymptotic level < 0.05 used as the set criterion for statistical significance and an area value of 1 indicating a strong/accurate prediction potential [33].

## 3. Results

At baseline, 58 girls completed all the measurements and were classified into two groups according to their menarche status as reported by the Status Quo method, namely, a pre-menarche (n = 13, 23%) and post-menarche group (n = 45, 77%). In the post-menarche group of 45 participants (77%) who had reached menarche at baseline, 8.8% (n = 4) reported reaching the milestone during Grade 5, 22% (n = 10) during Grade 6, 66.6% (n = 30) during Grade 7 and only 2% (n = 1) during their Grade 8 year. During the first (T2) and second (T3) follow-up measurements, 93% (n = 42, T2) and 98% (n = 54, T3) of all the subjects in the study had reached menarche. These statistics point to differences of up to five years between the times that subjects reached menarche first and late at a mean age of 15.51. One subject still had not reached menarche at the final measurements.

Table 1 reports the descriptive age characteristics of the pre- and post-menarche groups at baseline (February 2010) at the age of 13.51 ± 3.50 years, and again at the two follow-up measurements in February 2011 and February 2012 at the ages of 14.51 ± 3.51 and 15.51 ± 3.51 years, respectively. At baseline, no significant age differences (0.01 years, *p* > 0.05) were found between the pre- and post-menarche groups.

Four statistical methods were used to determine the classification agreement between the pre- and post-menarche groups, based on known actual menarche status. These include independent t-testing, a ROC curve analysis, cross-tabulation and logistic regression analysis.

Table 2 displays a comparison of the maturity offset values that were calculated for the pre- and post-menarche groups during T1, T2 and T3 by the maturity offset equation as developed by Mirwald et al. All categorization was conducted based on menarche status during T1, and only time point measures T1, T2 and T3 were used for further analysis.

Our results showed a strong realized power of 0.92 for the group in the first year of the study, (Table 2), which gradually declined at T2 and T3. This indicates that although the two menarche groups (pre- and post-) were small, there was still enough power to do the analysis during T1 (0.9185). However, the power of the analysis declined significantly during T2 and even more during T3. The mean values of each group as displayed in Table 2 are an indication of the years relative to the onset of PHV as compared to a score of 0.00 that represents the PHV.

In this regard, a score further away from 0.00 indicates the number of years the subject is pre-PHV (negative score) or post-PHV (positive score). At T1 during baseline, both groups showed positive values, indicating that they were post-PHV. The independent t-test showed that the pre-menarche group displayed a lower average maturity offset (+1.01 years) compared to the post-menarche group (+1.44 years), which agrees with their developmental status at that age. These differences between the groups were also statistical (*p* < 0.01) and of moderate practical significance (*d* = 0.73).

The maturity offset differences between the groups became smaller during T2 in the Grade 9 school year. Here the maturity offset scores of both groups moved further away from 0.00, with the pre-menarche group showing the largest change (+0.85 years) compared to +0.65 years in the pre-menarche group. These differences became statistical insignificant (*p* > 0.05), although it still showed small practical significance (*d* = 0.23). At T3, during the final measurements in Grade 10, the mean maturity offset values of both groups continued to move further away from 0.00. A mean group difference of 0.16 years was found, showing no statistical or practical significance (*p* > 0.05, *d* < 0.3).

These results were further analyzed by means of cross-tabulation by classifying the group into tertiles according to their maturity offset scores with the lower group representing late-maturing girls, the middle group representing average-maturing girls and the upper group representing early-maturing girls. The three tertiles were established by a 1-year (12 month) SD from the maturity offset scores. Table 3 reports the results of the cross-tabulation distribution of maturity offset values within known early-, average- and late-maturing groups as determined by the Status Quo method.

Regarding the pre-menarche group, (n = 13) who are expected to be distributed within the lower group, it was found that 61.51% were distributed within the lower maturity group during baseline measurements in Grade 8 at a mean age of 13.51 years. Furthermore, 53.85% (Grade 9) and 46.15% (Grade 10) of the pre-menarche group were distributed within the lower group at a mean age of 14.51 and 15.51 years, respectively. This showed poorer classification when moving further away from PHV, as can be seen by chi-square *p*-values that indicate a statistically significant back-classification during T1 (*p* = 0.042), whereafter no statistical significant back classification was found at T2 (*p* = 0.183) and T3 (*p* = 0.459). The results also revealed a larger classification of girls within the middle (average-maturing group) with increased age.

However, regarding to the post-menarche group, who are expected to be distributed within the upper group, only 35.56% were distributed within the upper group during baseline measurements, while 24.44% were wrongly classified into the lower group in Grade 8, 35.56% in Grade 9 and 2.22% in Grade 10. Again, the girls were less accurately classified at older ages, and the percentage classification was poorer compared to in the pre-menarche group.

The significance of these classification distributions during each measurement was assessed by means of Cramer’s V values. A Cramer’s V of 0.331 was found during the baseline measurements in Grade 8 that showed a moderate predictive effectiveness (0.3 < chi-square < 0.5). The significance of distributions, however, decreased with increased age as a Cramer’s V of 0.242 (small predictive effect) was found during Grade 9 and 0.164 (small predictive effect) in Grade 10.

Another analysis was performed by means of a ROC curve analysis to determine how well the maturity offset equation can distinguish between the pre- and post-menarche groups. Table 4 and Figure 1 reports the cut-off points that were determined by this method by plotting the true positive (menarche status) in the function of the false positive (maturity offset value) for different cut-off points and calculating the Youden index.

Each point on the ROC curve represents a sensitivity/specificity pair corresponding to a particular decision threshold. The area value (AUC) represents a measure of how well the maturity offset equation can distinguish between early- or late-developing subjects and post-menarche or early-developing subjects. The results indicated a predictive maturity offset value of 1.250, which can be used during the prediction process of predicting early- and late-maturing girls. In this regard, scores below the cut-off value (closer or further beneath 0) will be an indication of late-maturing girls (pre-menarche) and scores above the medium value (further away and above 0) will be an indication of early-maturing girls (post-menarche). The results in Table 4 and Figure 1 furthermore indicate an area value (AUC) of 0.757 where a score of 1 is indicative of accurate prediction, which shows that the prediction capability of the maturity offset equation is of relative high accuracy. Therefore, the process of classifying early- and late-maturing girls by means of the maturity offset equation is of statistical significance, as can be seen by the asymptotic significance of 0.005 (Table 4).

Table 5 displays the results of the cross-tabulation on the groups distribution of maturity values compared to the known pre- and post-menarche groups. These analyses revealed that the maturity offset equation could accurately predict 84.62% of the pre-menarche group into the early-maturing ROC group. Furthermore, the equation accurately predicted 66.67% of the post-menarche group into the late-maturing ROC group. This association was statistically significant and of medium effect (Cramer’s V = 0.43, *p* = 0.001).

Table 6 reports the results of the logistic regression as a fourth and final method to predict maturity status (Status Quo) from the maturity offset equation. During T1, both the Cox and Snell and the Nagelkerke R square values showed that the percentage variance explained is more than 10% (which can be considered as a medium effect) and that the maturity offset equation could predict menarche statistically significantly (*p* = 0.006) at T1. The odds ratio of 11.7 indicates that for every year further away from maturity offset, it becomes 11.7 times more likely that a girl would have reached menarche. Like previous analysis, this prediction was no longer statistically significant in subsequent years.

## 4. Discussion

The aim of this study was to investigate the level of accuracy between known self-reported age at menarche, or not reaching menarche status, and calculated maturity offset in pre- and post-menarche girls over a two-year follow-up period by means of an exploratory study. Four different statistical methods were applied to assess classification accuracy. Overall, all four methods that were used indicate the highest predictive power during the first year of the study, with reducing levels of accuracy when moving away from the baseline measurements. Furthermore, the results indicated strong realized power of 0.92 for the group in the first year of the study, indicating that although the two menarche groups (pre- and post-) were small, there was still enough power to classify accurately. However, the realized power gradually and significantly declined over time with increased age and moving further away from PHV. The maturity offset equation can therefore be considered as having the ability to determine differences in maturity offset with a degree of accuracy when compared to the actual maturity status. In this regard, results from cross-tabs indicate significant positive results during T1, where 61.51% were distributed correctly (*p* = 0.042) for the pre-menarche group, although it was inaccurate for the post-menarche group (35.56%, *p* > 0.05). Correct classification was the most significant and highest at younger ages and closer to PHV (T1, 13.51 years) although declining significantly with increased age beyond PHV.

The percentage of girls who have reached menarche during T1 (77%, 13.51 years), T2 (93%, 14.51 years) and T3 (98%, 15.51 years) (Table 1) coincides with other studies that found that girls on average reach menarche between 10 and 13 years of age, and approximately 6 to 12 months after peak height velocity (PHV) [21,34]. In agreement with this, various researchers also reported that menarche occurs on average at 12.8 ± 1.0 years in girls [15,16,17].

Due to practical reasons such as grouping girls during team sports or using this information during talent identification (TID), the importance of being able to classify girls as pre-, average or post-menarcheal or into different maturity groups as early or late developers, is relevant. Therefore, a non-invasive, valid, efficient, one-time assessment of biological growth and maturation of girls, which can be used in schools and field settings, is deemed important [18]. Although various versions of maturity offset equations to the original Mirwald equation have been developed to attempt increasing the accuracy and feasibility of these, our study used the original maturity offset equation that was developed by Mirwald and co-workers [18] to classify girls into different maturity groups, to determine the utility of such scores in our South African sample.

The values of the equation for girls in the pre- and post-menarche groups were in the correct order and increased over time, which showed that it has classification potential. The mean maturity offset differences in values of our group during Grade 8 (*p* = 0.00, *d* = 0.73) at a mean age of 13.51 years were of statistical and medium practical significance with both groups having positive maturity offset scores above zero. With increased age, pre- and post-menarche groups yielded non-statistically significant, between group differences in maturity offset values at T2 in Grade 9 at a mean age of 14.51 years (*p* = 0.11; *d* = 0.40) and at T3 in Grade 10 (*p* = 0.29; *d* = 0.27) at a mean age of 15.51 years (Table 2). These differences were of small practical significance only during T2. This phenomenon could be ascribed to the maturity offset equation that incorporates measures of the timing of growth including interactions and ratios that are related to earlier and later growth periods. As our group was already relatively late within their pubertal phase at baseline, during follow-up measures, most of these parameters would have been changed to a more mature state with increased age in all the girls. Although variability in biological growth among individuals of the same chronological age is large around the adolescent growth spurt and the onset of menarche occurs on average at 12.8 years of age [4], our mean group age were already 13.51 years at baseline (T1) with only 0.01 years’ difference between the pre- and post-menarche groups (Table 1). This higher mean age of the group compared to the mean age reported by [4] for reaching menarche worldwide could have resulted in differences in growth characteristics between early- and late-maturing girls already becoming insignificant during baseline measurements. This in turn may have influenced the maturity offset equation of the pre- and post-menarche groups at later chronological ages even more as they moved further away from their age of menarche. This is well aligned with more recent studies that also investigated amended version of the original Mirwald equations [15,16,17]. At older ages, growth differences are leveled out between early- and late-maturing girls and they show similar growth characteristics and ratios (this equation includes stature, weight, sitting height, leg length and interactions) that imply different stages of growth, which could result in more similar maturity offset values between groups. This finding is in line with the conclusions drawn from a study by Malina and co-workers where they compared the Mirwald and Moore equations, stating that the prediction equations may be potentially useful within one- or two-year CA groups, especially close to the time of PHV [17].

With regard to the maturity offset values that are displayed in Table 2, both pre- and post-menarche girls showed positive values during all measurements in Grade 8 (T1), 9 (T2) and 10 (T3), indicating that on average, all of the girls have reached PHV. PHV is a developing stage that occurs before reaching menarche, which is a very late marker in the puberty phase of girls [35]. This indicates that girls within the pre-menarche group already surpassed PHV (+1.01 years) and is indicative that they are within 1–2 years from reaching menarche, as menarche is reached 1–2 years after PHV [4]. These results agree with the menarche information of the subjects that showed that 76% (10 of the 13) of pre-menarche girls had reached menarche by the age of 15 years with only three (24%) being incorrectly classified. In this regard, caution should be raised when interpreting results from maturity offset equations, as various studies [15,16,17,18] concluded that when population specific equations are applied to other samples, there may be a loss in the accuracy of the equation. Mirwald and co-workers [18] recommended that when their original equation is used on other populations that it is standardized for, it should rather only be used as a classification tool. Within the current study, the equation under-predicted age at PHV where the results indicated that post-menarche girls should have reached PHV 1.44 years prior to baseline measurements (Table 2). These findings agree with results found when using the Moore equation [17]. The medical history of the post-menarche group indicated that 22 of the 45 post-menarche girls have already reached menarche before or at the age of 12 years, which is indicative that they would have reached PHV 1–2 years prior to 12 years of age. The post-menarche group (early developing girls), however, displayed maturity offset values that were further above 0.00 (Table 2) compared to pre-menarche girls (late-developing girls), which indicates that the equation does have potential as a classification tool. In this regard, the differences between pre- and post-menarche groups were statistically (*p* = 0.00) significant while also showing medium to large effect sizes (*d* = 0.73) during T1. It should, however, be noted that the equation should be used rather on an individual level and not for group classification, which might have contributed to these influences on the accuracy of the equation.

Furthermore, when dividing the group into tertiles, the pre-menarche (late developing girls) was distributed more accurately (46.15% to 61.54%) within lower maturity offset values during all measurements in Grade 8, 9 and 10 (Table 5) compared to the post-menarche group. Post-menarche or early-maturing girls were correctly distributed below 40% within the higher maturity offset values (Table 3). These results coincide with reported findings, indicating that the accuracy of the equation is more accurate within certain ages. Mirwald and co-workers [18] also suggested that their maturity offset equation is more accurate up to four years pre-PHV and no more than two years post-PHV and within the age span of 8 to 16 years. Although our groups fall within the set age range (8–16 years), a large percentage of the post-menarche group had reached PHV more than 2 years before the baseline measurements of the study, as was evident from our statistics showing that 37% of the post-menarche group reached menarche before or at 12-years of age, while it is known that PHV occurs 1–2 years prior to menarche. This might have contributed to the group of early developers being less accurately predicted or distributed into the upper group through cross-tabulation. It should, however, be emphasized that the results are based on mean values of the group that varied considerably in their menarche status within the broadly classified pre- and post-menarche groups that were compared. The analysis of the accuracy of the equation should rather be based on how accurate everyone with their own medical history could be classified based on this history.

Consistent results were also found when testing the ability of the maturity offset equation to accurately predict early- and late-maturing girls by ROC curve statistics. In this regard, an asymptotic significance value of 0.005 (Table 4, Figure 1) for predicting early- and late-maturing girls were found that was statistically significant in predicting maturity status with an area value of 0.757, which is close to a perfect prediction score of 1 with confidence intervals of 0.569 to 0.946 (Table 4, Figure 1). Our cross-tabulation analysis furthermore showed that 84.62% of pre- and 66.67% of post-menarche girls could be correctly classified into the ROC maturity groups (Table 5).

Logistic regression also indicates that the maturity offset equation can be used to predict menarche statistically significantly in these girls at age 13.5 years and that for every year further away from maturity offset, it becomes 11.7 times more likely that a girl would have reached menarche.

Our study therefore confirms that maturity offset as calculated by using an equation that is based on age and growth characteristics has potential for classification purposes within a South African population up to a mean age of 13.51 years. The results also indicate that the accuracy of the equation declines significantly from 13.5 years of age onwards in girls, which agrees with other findings in this regard. Based on the inaccurate classification of several of the participants, especially those who were the furthest in their pubertal development, the equation in its current form is considered to be used with caution in populations other than those it was developed for. More research is recommended in this regard to improve the utility of this equation by adjustments for specific population groups. In this study, the equation could unfortunately only be tested on white girls in SA, leaving the question whether the same results will be applicable to other ethnic groups living in South Africa. Currently, no studies on girls of this nature are available on SA girls, and studies to compare our findings with are furthermore only conducted on European samples [15,16,17]. More research is therefore needed in this area. However, our results do indicate that this equation has the potential to be used to broadly categorize girls earlier in the pubertal stage into maturational groups for research as well as for practical purposes. Furthermore, the results confirm that the accuracy of these equations declined with increasing age after PHV was reached. Maturity offset equations will also be more suitable for classification of females on an individual level, based on their own developmental history.

Although the study contributed to an improved understanding of using a maturity offset equation in girls by comparing it to actual menarche status that occurs later in the biological development of girls, especially in a South African sample of girls, the study had limitations that need to be taken into consideration when interpreting the findings. The data for this analysis were collected by means of a convenience sample that included only a small number of subjects, and mainly from one ethnic group (white, Caucasian girls). The study also formed part of a larger research project [36,37,38] and was only a sub-objective within this larger project. Consequently, although the group size was found to be large enough to have power for analysis, and also reached strong power of classification at T1, the results can only be generalized to a small extent and mainly to white, Caucasian girls in South Africa. In addition, although the data were collected over a follow-up period of two-years, which strengthens the findings that were obtained, the study could only focus on the time period between 13 and 15 years of age, which is the time period when children enter secondary schools in South Africa, which made it impossible to follow the group especially during an earlier age period as they enroll into different secondary schools after finishing primary school. A larger age-span period, especially from before the age of onset of PHV, is needed to fully investigate the usefulness and accuracy of maturity offset equations. The inclusion of a broader cultural diversity that is more representative of the South African girls is also strongly recommended.

Practitioners, physical educators, researchers and adolescent girls alike may benefit from a valid and once-off assessment of maturation that is measurable earlier than the age at menarche. In this regard, results from maturity offset equations will be useful to broadly categorize girls earlier into different maturity groups by practitioners, physical educators and researchers, and can also be used by adolescent girls to track and understand their current developmental level. This information can then be used by girls, together with their physical educators, to understand and manage their sport performance, participation in sport programs and maintaining PA levels before and after menarche, by adapting their PA and sport programs accordingly to the effects of menarche [36,37,38]. Improved understanding of their biological maturation will provide them with the necessary perspective of differences in comparison to their peers in their physical development and sport performance, and in turn will play a vital role in their psychological well-being, confidence and motivation for PA and sport participation, hence also addressing the affective domain of physical literacy. Further population and culturally specific research are therefore strongly recommended that can improve the use of these beneficial equations, to improve their usefulness. Although a small and convenient sample of girls was part of this study, it is, to our knowledge, the first of its kind being conducted on SA girls, making the results a unique contribution to knowledge in this area of South Africa, and more so from an African perspective, and therefore serves as a foundation for future research in this area within SA and Africa.

## 5. Conclusions

It is concluded that the current study makes a unique contribution to the understanding of the accuracy, effectiveness and usefulness of original or amended equations to determine maturity offset in South African girls. The findings of this study should, however, still be considered as exploratory in nature but foundational for future research on the usefulness and accuracy of maturity offset equations within a South African population of girls.

Although it is recognized that the results are based on a small and homogenic group, mostly focusing on white Caucasian girls, it can be concluded from all four statistical methods that were applied that the maturity offset equation as a proxy of growth differences in same aged girls does have value to categorize girls into early-, average- or late-maturing groups, especially when girls are close to or beyond PHV. However, the accuracy of calculating girls’ actual maturity offset, based on the equation, is not accurate. Furthermore, maturity offset equations seem to be more accurate and subsequently valid at a younger age, closer to PHV, which our results also confirmed, and it is therefore recommended that it should be used at younger ages closer to the PHV period.

Based on these findings, it is recommended that maturity offset equations should be used with caution when focusing on the actual maturity offset age of girls, and in turn these equations should be adjusted for each specific population regarding the growth characteristics of that population, which urges more research in this area. The physical literacy that girls can obtain through maturity offset results, however, does have value, especially regarding the current state and effect of menarche, which will provide them with a basic timeline of when they might reach menarche, the effects thereof on their PA and sport participation, and also the motivation to continue although expected setbacks related to their physical performance are anticipated.

## Figures and Tables

**Figure 1 jfmk-09-00127-f001:**
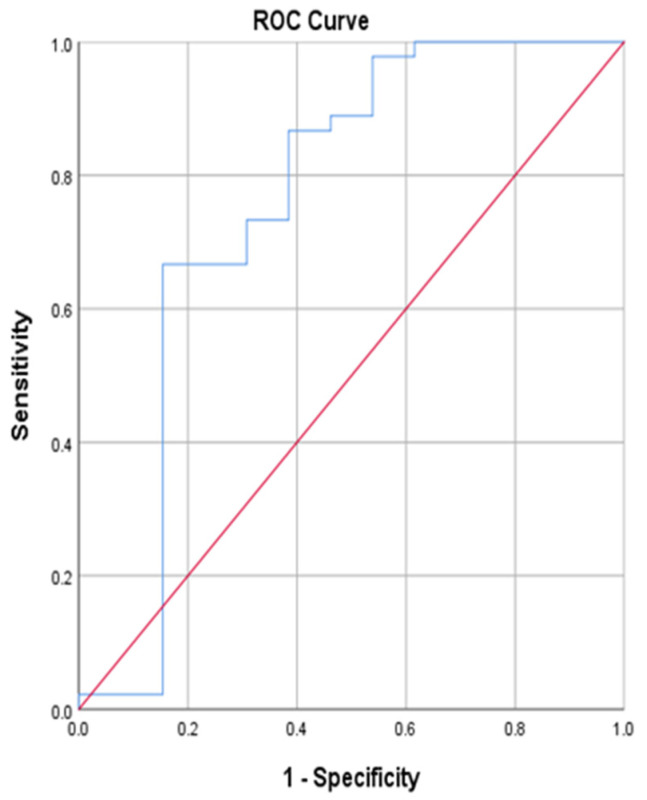
ROC curve of the maturity offset equation.

**Table 1 jfmk-09-00127-t001:** Descriptive age characteristics of the pre- and post-menarche groups.

	Year 1 (T1)		Year 3 (T3)	
	N	Mean Age ± SD	N	Mean Age ± SD
Group	58	(T1) 13.51 ± 3.50	58	(T3) 15.51 ± 3.51
Pre-menarche	13	(T1) 13.52 ± 3.58	13	(T3) 15.52 ± 3.58
Post-menarche	45	(T1) 13.51 ± 3.53	45	(T3) 15.51 ± 3.53

T1 = Year one (Grade 8); T2 = Year 2 (Grade 9); T3 = Year 3 (Grade 10); N = Group size.

**Table 2 jfmk-09-00127-t002:** Maturity offset of pre- and post-menarche girls from Grade 8 to Grade 10 and the statistical significance of differences between groups.

	Pre-Menarche			Post-Menarche			Significance of Difference		Realized Power
Variables	N	Mean [Min:Max]	SD	N	Mean [Min:Max]	SD	*p*-Value	*d*-Value	
Grade 8 (T1)	13	+1.01 [0.33:2.20]	0.59	45	+1.44 [0.74:2.27]	0.36	0.00 *	0.73 ^##^	0.9185
Grade 9 (T2)	13	+1.86 [1.25:2.96]	0.58	45	+2.09 [1.37:3.07]	0.41	0.11	0.40 ^#^	0.3579
Grade 10 (T3)	13	+2.59 [2.01:3.72]	0.58	45	+2.75 [1.77:3.77]	0.42	0.29	0.27	0.1922

N = Group size; SD = Standard deviation; T1 = Baseline measurements (Feb) in Grade 8; T2 = First measurements (Feb) in Grade 9; T3 = First measurements (Feb) in Grade 10; * = Statistical significance (*p* < 0.05); *d*-value = Practical significance; # = small practical significances (*d* > 0.2); ## = medium practical significance (*d* > 0.5); + = Predicted maturity as years post PHV.

**Table 3 jfmk-09-00127-t003:** Distribution of maturity offset values within known pre- and post-menarche girls from Grade 8 to Grade 10.

	n	Maturity Offset Grade 8			n	Maturity Offset Grade 9			n	Maturity Offset Grade 10		
		Lower Group	Middle Group	Upper Group		Lower Group	Middle Group	Upper Group		Lower Group	Middle Group	Upper Group
Pre-M	13	61.54%	23.08%	15.38%	13	53.85%	23.08%	23.08%	13	46.15%	53.89%	0.00%
Post-M	45	24.44%	40.00%	35.56%	45	26.67%	37.78%	35.56%	45	28.89%	68.89%	2.22%
Chi-square		0.042				0.183				0.459		
Cramer’ Vs		0.331				0.242				0.164		

n= Group size; % = Percentage distribution within each group; Chi-square > 0.1 = Small effectiveness; Chi-square > 0.3 = Medium effectiveness; Chi-square > 0.5 = Strong effectiveness.

**Table 4 jfmk-09-00127-t004:** Cut-off point for maturity offset to classify pre- and post-menarche girls.

Area	Std. Error	Asymptotic-Significance	Predicted Cut-Off	Asymptotic 95% Confidence Interval	
				Lower Bound	Upper Bound
0.757	0.096	0.005 *	1.250	0.569	0.946

“Area” closer to 1 = More accurate classification/prediction; * = Statistically significant (*p* < 0.05).

**Table 5 jfmk-09-00127-t005:** Classification of maturational groups according to the cut-off points of the ROC curve analysis.

	Late-Maturing ROC Group	Early-Maturing ROC Group
Pre-menarche group	84.62%	15.38%
Post-menarche group	33.32%	66.67%
Chi-square *p*-value	0.001	
Cramer’s V	0.430	

**Table 6 jfmk-09-00127-t006:** Logistic regression analysis of the ability of the maturity offset equation to accurately predict maturity status.

	B	S.E.	Wald	df	*p*-Value	Odds Ratio	95% C.I. for Odds Ratio			
							Lower	Upper	Cox & Snell R Square	Nagelkerke R Square
T1	2.461	0.891	7.624	1	0.006	11.718	2.042	67.228	0.156	0.238
T2	1.197	0.759	2.489	1	0.115	3.310	0.748	14.639	0.046	0.070
T3	0.765	0.720	1.127	1	0.289	2.148	0.523	8.814	0.020	0.031

B = estimated coefficient; S.E. = standard error; df = number of degrees of freedom for the model.

## Data Availability

Data are available from the principal investigator of the study if or when required.

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
