# Peer review of "Exploratory Study into the Classification Agreement between Self-Reported Age of Menarche and Calculated Maturity Offset in Adolescent Girls: A Two-Year Follow-Up Study"

_jfmk, 2024, doi:10.3390/jfmk9030127_

Round 1

Reviewer 1 Report

Comments and Suggestions for Authors

Dear Authors, 

thank you for the possibiliy you gave me to revise this manuscript. Here there are my advices to improve the impcat of the manuscript

Manuscript Title:

Considering the poor sample size, I will suggest you to switch your study into an exploratory or pilot study and to indicate this into the title

Abstract

It is ok

Introduction:

From line 29 to 39 you provide completely useless information with no connection with the other part of the text. Please eliminate this part and start your introduction directly from line 40. 

Despite this, the introduction is comprehensive and well-structured, presenting relevant background information and clearly stating the study's purpose. It provides a solid rationale for the research by discussing the importance of understanding menarche and maturity offset in adolescent girls.

Methods:

You declared to have included 58 girls. How did you calculated the sample size? I beleave that this sample is too small to support your findings, considering that you splitted the sample into four groups to get more inferences. Therefore, I suggest you to turn this study into an exploratory one. In addition, this limitation is not adequately addressed in the manuscript, which raises concerns about the statistical power and robustness of the findings.

Results:

The results are presented clearly, with appropriate use of statistical analyses to compare the classification agreement between self-reported menarche and calculated maturity offset. 

Discussion:

The discussion appropriately interprets the findings in the context of existing literature and highlights the study's contributions to understanding maturity offset and menarche classification. However, it fails to critically evaluate the impact of the small sample size on the study’s conclusions. A thorough discussion on how the sample size limitation might have influenced the results, including the potential for type II errors, is necessary for a balanced and credible interpretation of the findings.

Conclusion:

conclusion is ok

Finally, I am a non-native English author. Therefore, even if I do not have detected troubles in Use of English, I will suggest you, if you do not have still done yet, a revision from a native English author, to ensure a proper and correct Use of English. 

Author Response

Good morning reviewer

Thank you for all your comments and suggestions. Please find attached the feedback to all your comments and suggestions. You are reviewer 1 in this document.

Kind regards

Reviewer 2 Report

Comments and Suggestions for Authors

Dear authors,

Congratulations on your work on a non-invasive surrogate for classifying the maturity status of girls, an alternative to menarche. The study is significant as it can facilitate the collection of this sensitive variable in research protocols. However, I identified some points that deserve attention when reviewing the manuscript, starting with the abstract. I request a response letter addressing each point below, along with the updated version of the manuscript with the changes highlighted.

Best regards,

The Reviewer

------------------------------------------------------

ABSTRACT

1. In general, the abstract needs greater objectivity. The objective of the study was not presented. I suggest going into more detail about the methods and including statistical parameters in the results.

2. Conclusions must respond directly to the research objectives. Furthermore, remove instructions to authors present in the template, such as: "(4) Conclusions: indicate the main conclusions or interpretations. The abstract should be an objective representation of the article and it must not contain results that are not presented and substantiated in the main text and should not exaggerate the main conclusions."

INTRODUCTION

3. The introduction needs to be more direct and objective, highlighting only the essential points that contextualize and justify the research, without going into unnecessary details that make the text tiring. I recommend that authors try to clearly explain the importance of physical literacy without delving into specific domains.

4. Present directly how menarche affects girls' participation in physical activities and sports.

5. Please, mention the main limitations of current methods for determining biological maturity.

6. Reduce the number of unnecessary references in the introduction to make it more attractive and relevant.

7. In the last paragraph, justify the need for non-invasive methods to assess biological maturity and conclude the introduction by presenting the objective of the study.

METHODS

The most critical point in the methods described is the lack of clarity and standardization in the description of anthropometric measurement procedures. Organize the methods so that they are better understood, without excessive details, so that they can be replicated by other research.

8. It may be necessary to simplify several sentences, as the text seems truncated and difficult to understand. For example: "This study made use of a longitudinal research design. The main project, 'Growth and sport psychological characteristics of talented adolescents', was performed over a 3-year school period (2010-2012)." That could be “This longitudinal study, 'Growth and sport psychological characteristics of talented adolescents', spanned 2010 to 2012.”

9. Your sample appears to be very small, which is acceptable for the study design. But it is important to provide a justification for the sample size, including the power of the test (a posteriori) to detect significant differences between groups.

10. The methods section is not objective and lacks conciseness at several points. I recommend moving the details about frequency of measurements to a subsequent section on procedures, keeping only the information in the first topic.

11. Please eliminate or consolidate repetitive information about age groups over the years.

12. There are many details about the description of some variables, for example, height. Likewise in relation to weight. Remove overly detailed descriptions that don't directly impact results.

13. What test was used to analyze the normality of the data? Was there a normal distribution? If there wasn't, why did they use parametric tests?

RESULTS

14. Some tables need to be better prepared (for example Table 6). See how to improve data presentation.

15. It is also necessary to optimize the description of results in shorter sentences. The initial part (for example) where the pre- and post-menarcheal groups are presented, the information should be structured in more objective sentences highlighting the key numbers and percentages.

DISCUSSION

Authors need to be more objective in the discussion. There are long paragraphs and long sentences to say something simple. This is in almost the entire text. Please review the manuscript.

16. The first paragraph is unnecessary. You can even start by recalling the objective of the study, but you should mainly highlight the main results found and their practical usefulness.

17. Remove any words “Tables” from the discussion. Also remove all information about statistical tests from the section. If you can't remove them all, leave as little as possible.

18. The authors mention that the results can only be generalized to white Caucasian girls in South Africa due to the limited sample and ethnic homogeneity. How to increase the robustness of the study? Include information about how these results may be applicable to other ethnic and geographic populations.

CONCLUSIONS

19. Be more direct when responding to the general objective of the study. Overall, the conclusion is extensive and could be more succinct.

20. The authors mention the need for adjustments for different populations. What specific strategies for validating the equations in more ethnically and geographically diverse populations can be mentioned?

Author Response

Good morning reviewer

Thank you for all your comments and suggestions. Please find attached the feedback to all your comments and suggestions. You are reviewer 2 in the rebuttal. 

Kind regards

Reviewer 3 Report

Comments and Suggestions for Authors

Dear authors, the study is interesting, but it would be more interesting if they compared their results with much more recent research. Much of the approach taken in the introduction is based on very old literature. If the aim is to establish current methodological fields, it is not justified to rely on possibly obsolete studies dating back to 1974 (e.g., lines 73-75; 76-94; 101-116).

Also, in the abstract, review the conclusions (lines 22-25).

Please indicate the ethnic origin of the participants.

The age classification of the participants is very confusing. Please clarify the ages reflected in lines 162 and 167. Only one age should be indicated (The whole year was the midpoint of the range defining). Lines 168 to 173 are dispensable because all participants are one year older each season. Therefore, only column 1 should be shown in table 1.

Rather, the number of participants should be modified according to the number of participants reaching menarche in each year.

Lines 98 to 204 should be in the results.

The results are based on the use of a prediction equation from (Mirwald et al., 2001, but since then these equations have been updated a lot It is recommended to update the data according to Koziel (2018), Malina (2021), Moore /2015) among many others. 

Update the discussion with more current literature references, e.g. reference is made to Malina 2001 who has published new research in 2021 onwards. 

Please summarize the conclusions and indicate limitations of your research.

Author Response

Good morning reviewer

Thank you for all your comments and suggestions. Please find attached the feedback to all your comments and suggestions. You are reviewer 3 in the rebuttal. 

Kind regards

Round 2

Reviewer 1 Report

Comments and Suggestions for Authors

Dear authors,

Thank you for your effort in improving the manuscript. Your paper has consistently improved. 

Nice job!

Reviewer 2 Report

Comments and Suggestions for Authors

Dear authors,

Congratulations on submitting the improved manuscript.

Your efforts in making the requested adjustments are noticeable.

To make the manuscript even better, I suggest once again reviewing the introduction and conclusion to optimize the text, which is still long. Conclusions, especially, need to be more direct. Therefore, as an important recommendation, I ask you to review these two sections to make the text more objective and direct.

Yours sincerely,

The Reviewer.

Reviewer 3 Report

Comments and Suggestions for Authors

Dear authors, I believe that the modifications made are in line with the methodology and results obtained. From my point of view, the final document is clearer and I thank you for the effort made to clarify the aspects indicated in round 1. Finally, the conclusions have been profoundly modified and are better supported by the results.

Best of luck for the future.